# Ionosonde Observations of Spread F and Spread Es at Low and Middle Latitudes during the Recovery Phase of the 7–9 September 2017 Geomagnetic Storm

Lehui Wei [1], Chunhua Jiang [1,*], Yaogai Hu [1], Ercha Aa [2], Wengeng Huang [2], Jing Liu [3], Guobin Yang [1] and Zhengyu Zhao [1]

1 School of Electronic Information, Wuhan University, Wuhan 430072, China; lehuiwei@whu.edu.cn (L.W.); yaogaihu@whu.edu.cn (Y.H.); gbyang@whu.edu.cn (G.Y.); zhaozy@whu.edu.cn (Z.Z.)
2 National Space Science Center, Chinese Academy of Sciences, Beijing 100190, China; aercha@nssc.ac.cn (E.A.); huangwg@nssc.ac.cn (W.H.)
3 Institute of Earthquake Forecasting, China Earthquake Administration, Beijing 100036, China; liujingeva@163.com
* Correspondence: chuajiang@whu.edu.cn

**Abstract:** This study presents observations of nighttime spread F/ionospheric irregularities and spread Es at low and middle latitudes in the South East Asia longitude of China sectors during the recovery phase of the 7–9 September 2017 geomagnetic storm. In this study, multiple observations, including a chain of three ionosondes located about the longitude of 100°E, Swarm satellites, and Global Navigation Satellite System (GNSS) ROTI maps, were used to study the development process and evolution characteristics of the nighttime spread F/ionospheric irregularities at low and middle latitudes. Interestingly, spread F and intense spread Es were simultaneously observed by three ionosondes during the recovery phase. Moreover, associated ionospheric irregularities could be observed by Swarm satellites and ground-based GNSS ionospheric TEC. Nighttime spread F and spread Es at low and middle latitudes might be due to multiple off-vertical reflection echoes from the large-scale tilts in the bottom ionosphere. In addition, we found that the periods of the disturbance ionosphere are ~1 h at ZHY station, ~1.5 h at LSH station and ~1 h at PUR station, respectively. It suggested that the large-scale tilts in the bottom ionosphere might be produced by LSTIDs (Large scale Traveling Ionospheric Disturbances), which might be induced by the high-latitude energy inputs during the recovery phase of this storm. Furthermore, the associated ionospheric irregularities observed by satellites and ground-based GNSS receivers might be caused by the local electric field induced by LSTIDs.

**Keywords:** spread F and spread Es; irregularities; atmosphere gravity waves; geomagnetic storm; electrodynamic coupling

## 1. Introduction

Spread F phenomena have been extensively investigated for several decades since they were first observed by Booker and Wells [1] using ionosondes. Ionospheric irregularities or spread F can be observed frequently by ionosondes associated with scattered echoes traces in ionograms [2]. In the first instance, investigators have noted that spread F often appeared in equatorial regions, where it is well known as equatorial spread F/irregularities, which are typical characteristics of the nighttime ionosphere. Moreover, spread F can also be observed over low-latitude regions (divided in a standard of geographic latitudes in this paper). Different physical mechanisms have been put forward to reveal spread-F and its physical mechanisms [3–6]. Among these, the Generalized Rayleigh–Taylor (GRT) instability [7] is the most typical disturbance source for nighttime spread F at equatorial and low-latitude regions by diverse physical mechanisms, including atmosphere gravity

waves, neutral wind, magnetic and electric fields, together with background ionospheric conditions [5,8,9]. Under quiet electromagnetic disturbance conditions, the prereversal enhancement (PRE) of the electric field can accelerate the speed of the upward vertical plasma drift. The increased vertical drift will cause the height of the F layer in the equatorial and low-latitude regions to rapidly rise, then lead to a substantial increase in the growth rate of the Rayleigh–Taylor instability [4]. In other words, there is a widespread belief that the PRE provides a favorable condition for GRT instability formation in the equatorial and low-latitude region, causing the F layer to be lifted to a higher altitude and its electron density gradient to become steeper. The R-T instability growth rate enhances, and irregularities associated with plasma bubbles are triggered in the bottom side of the ionosphere and evolve from the height of the ionospheric F peak to the topside ionosphere in a nonlinear way [4,5,8,10]. It should be mentioned that, as a possible seeding mechanism, atmosphere gravity waves (AGWs) also play a significant role in the formation of equatorial spread F (ESF). The fluctuations in atmosphere density and temperature induced by gravity waves would alter the recombination rate of the ionospheric plasma and create the initial seed ionization disturbances (wave-like electron density perturbations) at the bottom side of the F layer. Such a "seed process" can initiate and amplify the R-T instability resulting in the formation of large-scale plasma irregularities and plasma bubbles [11,12].

Since the 1970s, mid-latitude spread F (MSF) has attracted much investigation ([2,3] and references therein). There is a type of MSF caused by the equatorial plasma bubbles and irregularities. The equatorial ionospheric irregularities in the F layer can extend to the middle latitude region along the magnetic field lines [13]. Another type of spread F is the locally generated irregularities at middle latitudes. The GRT instability is not efficient due to the larger geomagnetic dip angle at middle latitudes. Instead, Perkins [14] proposed the local Perkins instability might play a significant role in the formation of spread F at middle-latitudes. However, without the seeding sources in the ionosphere at middle latitudes [15], the growth rate of the Perkins instability is extremely small [16]. Hence, some researchers [3,15] suggest AGWs/traveling ionospheric disturbances (TIDs) are more responsible in the formation of spread F. Another theory based on coupled E and F region electrodynamics when the sporadic E layers occur was also put forward to demonstrate midlatitude spread F at nighttime, which is an effect of the Perkins instability [17–19]. Tsunoda and Cosgrove [18] indicated that E and F region irregularities structures at nighttime could be connected by magnetic field lines.

Although many investigations have widely explained various aspects of spread F and associated plasma irregularities, the storm-time variability of spread F and plasma bubble irregularities are very complicated and have not been understood comprehensively so far. Under geomagnetically disturbed conditions, Kelley et al. [20] and Buonsanto [21] suggested that the convection electric field can penetrate from the magnetosphere into the ionosphere due to its ineffective shielding by ring current charges. The direction of the convection electric field is eastward (westward) during the daytime (nighttime), as indicated by Kelley et al. [20]. The prompt penetrating electric field (PPEF) at low latitudes usually occurs during the main phase of a geomagnetic storm [22,23]. On the other hand, during a geomagnetic storm, Joule heating occurs at high latitudes, which causes an intense disturbed global thermospheric neutral wind circulation and then initiates the disturbance dynamo electric field (DDEF) in the ionosphere [24]. DDEF frequently occurs a few hours to few days after the beginning of the storm. Although sometimes DDEF turns from westward to eastward around 21:00–22:00 LT [25], it is normally westward in the evening. Westward DDEF at nighttime could suppress the growth rate of the R-T instability and consequently inhibit the formation of ionospheric irregularities during the recovery phase of the storm [26].

The 7–9 September 2017 magnetic storm, which had a double main phase, was one of the most intense storms of the 24 solar cycle. The strongest flare (class X9.3) of the current solar cycle, occurred on 6 September 2017. A coronal mass ejection (CME) event associated with the flare, was triggered and caused disturbed magnetospheric and ionospheric condi-

tions [27–29]. Many studies have analyzed and described the characteristics and behavior of ionospheric irregularities during this storm [30–35]. Recently Atıcı et al. [32] investigated the occurrence of irregularities during this storm at all latitudes using the international GNSS System (IGS)-GPS network. They concluded that this storm caused more ionospheric irregularities in the southern hemisphere than in the northern hemisphere, especially at high latitudes. Jin et al. [34] investigated the characteristics of the field-aligned ionospheric irregularities at low latitudes using a Hainan coherent scatter phased array radar and a Digisonde. Moreover, Aa et al. [30,31] studied postsunset midlatitude plasma bubbles over both American and Asian sectors during the first and second main phase of this geomagnetic storm on 7–8 September 2017. There is no doubt that the ionospheric irregularities associated with the main phase of the storm have been widely studied. However, to the authors' knowledge, the formation of ionospheric irregularities during the storm recovery on 9 September 2017 has not yet been observed and investigated. In this work, multiple observations are used, including ionosondes, Swarm satellites, and ground-based GNSS observations, to focus on the development process and evolution characteristics of the nighttime spread F/ionospheric irregularities and spread Es at low and middle latitudes during the recovery phase of the geomagnetic storm on 9 September 2017. This study can enrich the plasma irregularities cases over this geomagnetic storm and provide another view to study ionospheric irregularities over western China. Available for this study is a meridionally aligned (~100°E) chain of three ionosondes (PUR, LSH, and ZHY) that can be collectively used to study the latitudinal variation of ionospheric irregularities. Therefore, this study can gain a comprehensive understanding of the evolution and associated mechanism of spread F at low and middle latitudes during the whole geomagnetic storm.

## 2. Instruments and Data Processing

In this study, ionosondes, Swarm satellite measurements, and ground-based Global Navigation Satellite System (GNSS) ionospheric total electron content (TEC) observations were used to investigate nightside spread F characteristics during the recovery phase of the 7–9 September 2017 geomagnetic storm.

A meridional chain of ionosondes was installed at Puer (PUR, 22.7°N, 101.05°E, dip latitude 12.9°N), Leshan (LSH, 29.6°N, 103.74°E, dip latitude 19.8°N), and Zhangye (ZHY, 39.4°N, 100.13°E, dip latitude 29.7°N), along the longitude of about 100°E. As shown in Figure 1, red dots represent the geophysical distribution of these ionosondes. The ionosonde system can carry out a range of vertical incidence ionospheric sounding from 2 to 20 MHz and routinely operates every 5 min. Furthermore, an advanced software tool ionoScaler, created by Jiang et al. [36–38], was used to obtain automatic and manual scaling of the ionograms. This software tool was used to manually extract the virtual base height of the F2 layer (h'F2), the critical frequency of the F2 layer (foF2), and traces of spread F in ionograms to study variations of ionospheric irregularities.

In addition to ground-based ionosonde observations, measurements from the Swarm satellites were also analyzed to study the spread F and ionospheric irregularities during the recovery phase of this storm. Swarm is a constellation mission of the European Space Agency (ESA) [39], including three identical low Earth orbiting satellites named Swarm A, Swarm B, and Swarm C. They were launched on 22 November 2013. Langmuir probes were equipped on the Swarm satellites to obtain in-situation measurements of the electron density in the ionosphere. Swarm A and C fly side by side at an altitude of about 470 km; Swarm B flies at a higher altitude of about 530 km. Swarm A made several passes along a longitude of about 100°E on 9 September 2017. The red triangles in Figure 1 indicate the moving direction of the Swarm A satellite on 9 September. It provides an opportunity to further analyze ionospheric irregularities during the recovery phase of this storm from the topside perspective.

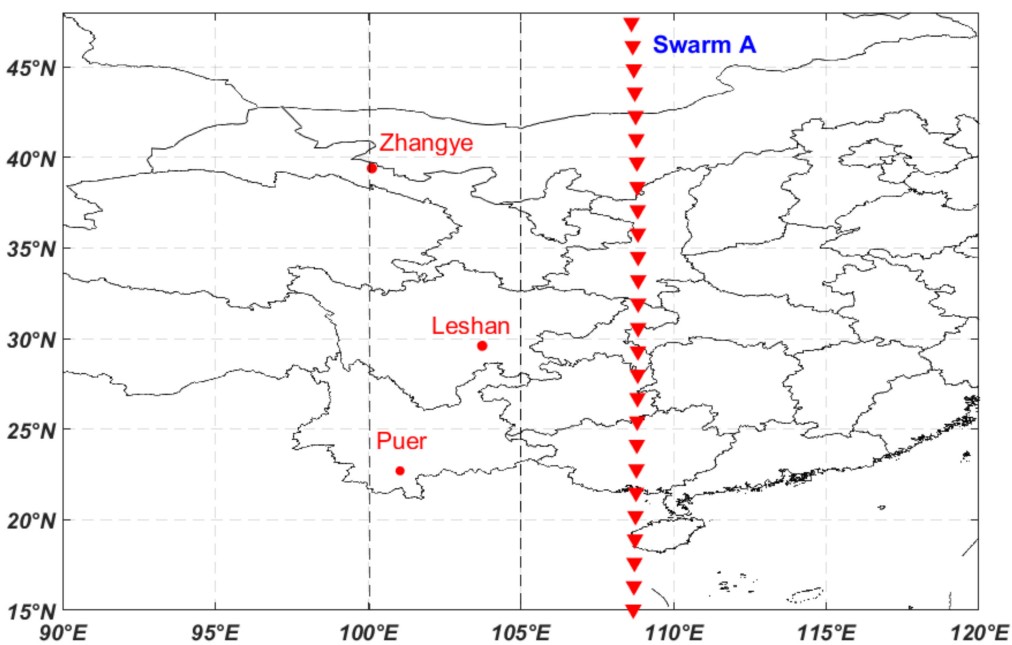

**Figure 1.** Red dots represent the geographic distribution of the three ionosondes (PUR, LSH, and ZHY considered in this study). Red triangles indicate the moving direction of the Swarm A satellite during the recovery phase of the storm on 9 September 2017.

GNSS receiver networks, including the International GNSS Service (IGS) and Crustal Movement Observation Network of China (CMONOC) can monitor ionospheric variability continuously [40]. In this study, the rate of TEC index (ROTI) defined by Pi et al. [41] was used to study ionospheric irregularities.

## 3. Observations

Figure 2 presents the geomagnetic and interplanetary conditions (top 4 panes) and the variations of ionospheric parameter h'F2 (bottom 3 panes) obtained from ionosondes data over PUR, LSH, and ZHY for 6–9 September 2017. The red vertical dashed lines indicate the start time of two SSCs. The black vertical dashed lines indicate the onset time of the second main phase and recovery phase of the storm.

The geomagnetic and interplanetary conditions were examined using planetary K index (Kp), disturbance storm time index (Dst), the north–south component of the interplanetary magnetic field (IMF-Bz), and the auroral electrojet (AE) index. The geomagnetic field indices (Kp, Dst, and AE) and IMF Bz, during the storm of 6–9 September 2017 are shown in Figure 2a–d, respectively. A CME event associated with the flare caused two geomagnetic storms on 7–8 September 2017. It can be seen from the geomagnetic indices, especially for Dst and AE indices in Figure 2b,d. According to the Dst, two sudden storm commencements (SSC) occurred at around 23:43 UT on 6 September and 23:00 UT on 7 September 2017. After the first SSC, IMF-Bz turned northward around 00:25 UT on 7 September with a maximum of ~15 nT. The second SSC started with a northward IMF-Bz with a maximum of ~32 nT. That marked the beginning of the second magnetic storm. The second main phase of the storm occurred between ~12:00 UT and ~20:30 UT on 8 September. After that, it was time for the recovery phase of the storm.

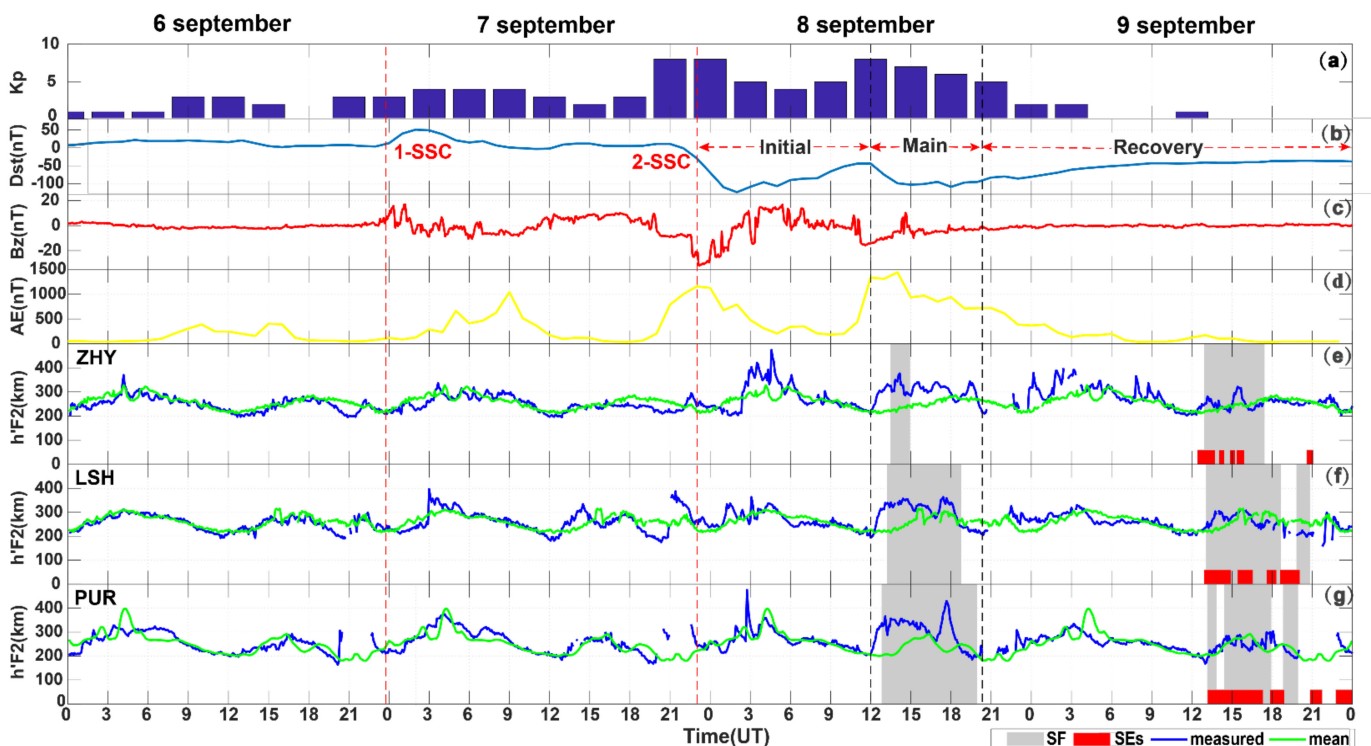

**Figure 2.** Interplanetary and geomagnetic conditions for the period of 6–9 September 2017: (**a**) Kp index; (**b**) Dst index; (**c**) IMF Bz; (**d**) AE index. The h′F2 at the ionosondes for 6–9 September 2017 (marked as blue lines): (**e**) PUR; (**f**) LSH; (**g**) ZHY. The average values of the h′F2 (2–6 September 2017) are shown as green lines. The red vertical dashed lines indicate the start time of two sudden storm commencements (SSCs). The black vertical dashed lines indicate the onset time of the second main phase and recovery phase of the storm. The horizontal gray bars and red bars indicate a period when spread F and spread Es occurred, respectively.

In Figure 2e–g, the h′F2 at the ionosondes for 6–9 September 2017, are measured and color coded as blue lines. The average value of the h′F2 on quiet days 2–6 September 2017 were plotted as green lines. Figure 2e–g also demonstrate the existence of spread F during the main phase and the recovery phase of the second Dst depression. The horizontal gray and red bars indicate a period of the existence of spread F and spread Es, respectively. These gaps between the gray bars and red bars in Figure 2e–g indicate there is no spread F or no spread Es. It should be noted that spread Es and spread F can be simultaneously observed only during the recovery phase of the magnetic storm. Specifically, during the second main phase of the storm, the h′F2 recorded at three stations increased almost simultaneously at ~12:00 UT (19:00 LT) on 8 September, and then reached the first peak altitude of ~376 km (ZHY) at 14:10 UT, ~356 km (LSH) at 14:25 UT, and ~354 km (PUR) at 14:20 UT, respectively. That is remarkably higher compared to its typical quiet day enhancement during local sunset hours. Then 45 to 80 min after the h′F2 enhancement onset, the ionograms at PUR, LSH, and ZHY sequentially show strong spread echoes. Furthermore, the duration times of spread F observed from ionograms at PUR, LSH and ZHY are about 12:45 UT-19:55 UT (~7 h), 13:15 UT-18:40 UT (~5.5 h), and 13:20 UT-14:50 UT (~1.5 h) during the second main phase of the storm on 8 September, respectively. The higher the latitudes spread F occurred at, the shorter the duration time of spread F is. It should be mentioned that spread Es was not observed during the second main phase of the storm. In the recovery phase period, the h′F2 disturbance of three ionosondes on 9 September gradually leveling off, but still there are some slight disturbances compared with the average value of the h′F2 on quiet days. The h′F2 of three stations also starts to rise at 12:20 UT (ZHY), 12:20 UT (LSH) and 13:05 UT (PUR), respectively. However, the amplitude of the h′F2 uplift is not as drastic as during the second main phase of the storm. Then spread F first occurred at ZHY at 13:05 UT, LSH

at 13:15 UT, and PUR at 13:20 UT. The onset time of spread Es is 12:25 UT (ZHY), 13:05 UT (LSH) and 13:15 UT (PUR), respectively. It can be seen from Figure 2 that the duration of spread F and spread Es is not consistent and not completely continuous.

Figures 3–5 show the evolution of spread-F on 9 September 2017 at ionosondes of PUR, LSH and ZHY, respectively. The satellite traces (STs) of the F layer are indicated by blue arrows. The oblique traces (OTs) of the Es layer are marked as red arrows.

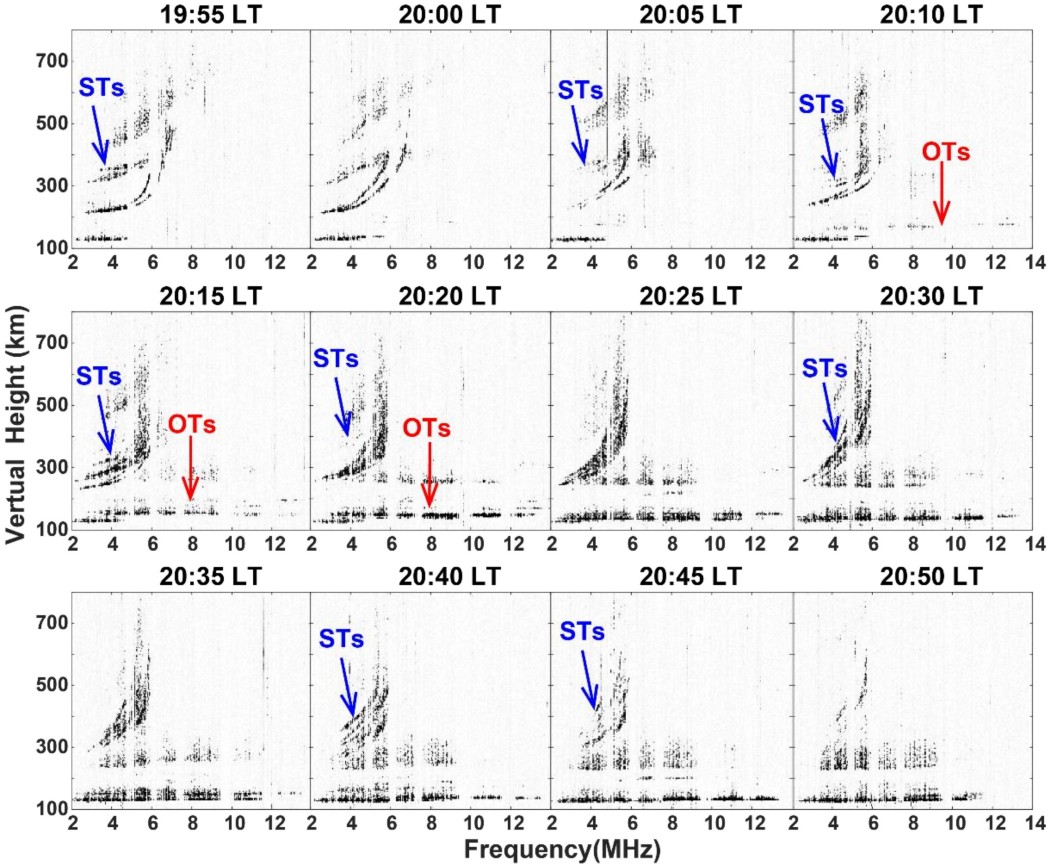

**Figure 3.** Evolution of spread-F at Puer (PUR) ionosonde during 19:55~20:50 LT (12:55~13:50 UT) on 9 September 2017.

Figure 3 shows the satellite traces [42] with a 1F doublet at PUR station are clearly seen at 19:55 LT. The satellite trace is separated from the vertical trace of the main F layer by about 100 km virtual range and then moves downward and moves closer to the main F layer trace over time. After about 10 min, the vertical echo trace of the F layer overlapping with the satellite trace begins to expand to higher altitudes. The onset of ionospheric irregularities is marked by an expansion in the vertical traces of ionograms and hence the virtual height of F layer traces becomes ambiguous. Diffuse echoes of the F layer completely covered F region traces on ionograms until 20:20 LT. The oblique traces of the Es layer persists in the ionograms at 20:00 LT and a slight spread in the oblique echo starts at 20:15 LT, accompanied with the considerable spread Es layer second hop traces. Similarly, the vertical echoes and oblique echoes of the Es layer are subsequently entangled at 20:25 LT and a clear spread can be seen in Es layer traces with maximum foEs of 13 MHz.

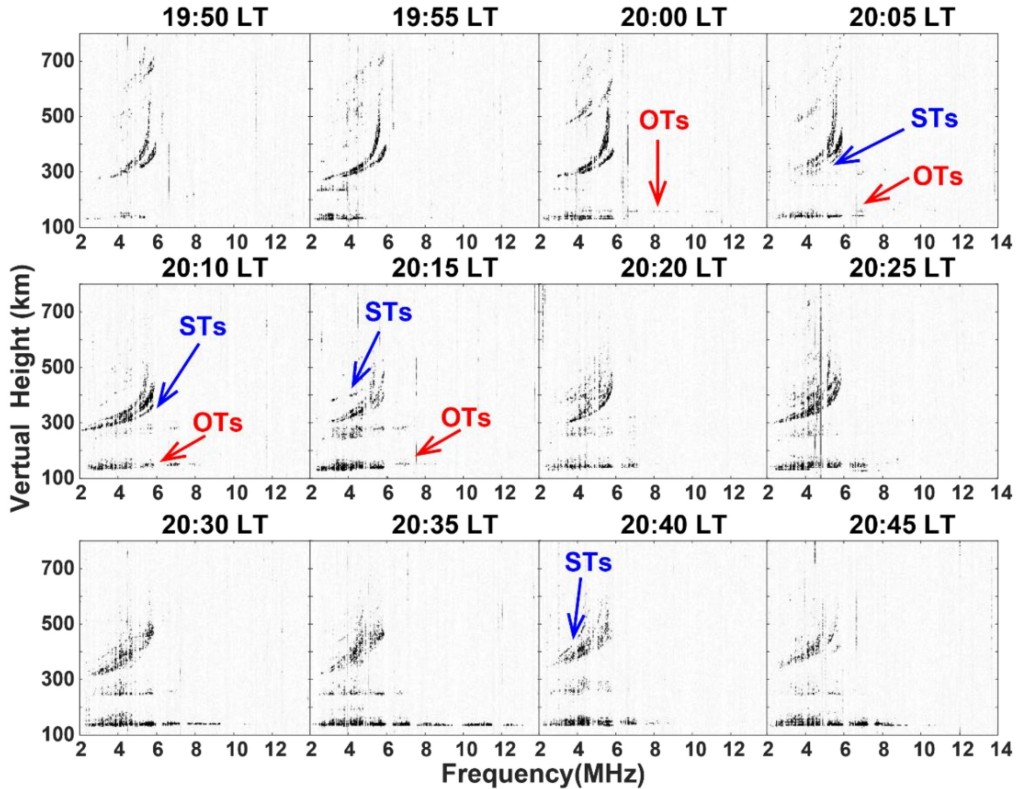

**Figure 4.** Evolution of spread-F at Leshan (LSH) ionosonde during 19:50~20:45 LT (12:50~13:45 UT) on 9 September 2017.

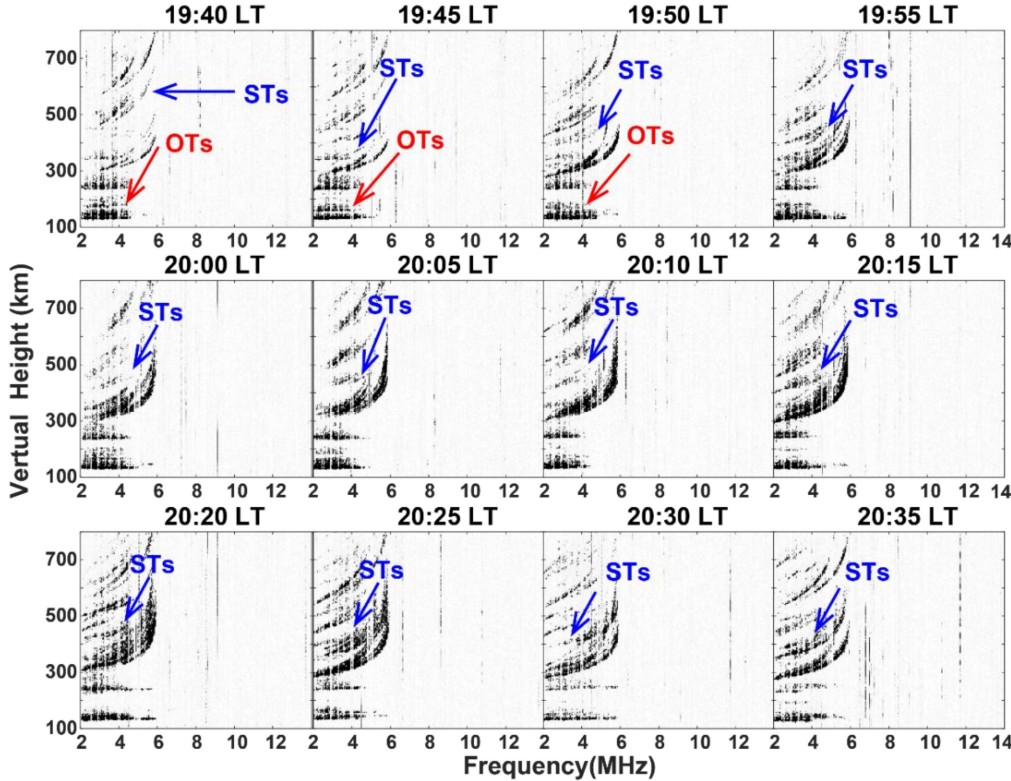

**Figure 5.** Evolution of spread-F at Zhangye (ZHY) ionosonde during 19:40~20:35 LT (12:40~13:35 UT) on 9 September 2017.

Figure 4 shows the visual variations of spread echoes on ionograms for the LSH station. Weak oblique trace and satellite traces were observed near their respective main

trace, the onset time is 20:00 LT and 20:05 LT, respectively. Same as Figure 3, the vertical traces and oblique traces of F region and Es layer start to merge at 20:05 LT, and gradually diffuse in several main traces for a period of time. At 20:15 LT, the complete spread F and spread Es are seen. During this period, second hop traces associated with F and Es layers show up from time to time and manifest modest scatter.

Another ionosonde located at ZHY is presented in Figure 5, which shows a series of ionograms taken during 19:40~20:35 LT (12:40~13:35 UT). The first ionogram shows the replicas of the usual F layer traces (as indicated by black arrowhead), which exists between the virtual height of first hop (1F) and second hop (2F) at 19:40 LT. Spread Es and STs always existed between 19:40 LT and 20:35 LT. The complete spread F and spread Es are seen at 20:15 LT.

Figure 6 shows the virtual heights at various frequencies (from 2 to 14 MHz with a step of 0.05 MHz) at ZHY, PUR, and PUR stations on 9 September 2017(11:00–21:00 UT). The black lines indicate the downward movement of the phase velocity of TIDs in the ionosphere, and gray bars represent the duration of spread F. The red arrows are used to mark the TIDs fluctuation period. As shown in Figure 6, the rough period of TIDs are ~1 h at ZHY station, ~1.5 h at LSH station and ~1 h at PUR station, respectively. Due to the limitations of observation equipment, the wavelength and propagation speed of the TIDs cannot be accurately calculated.

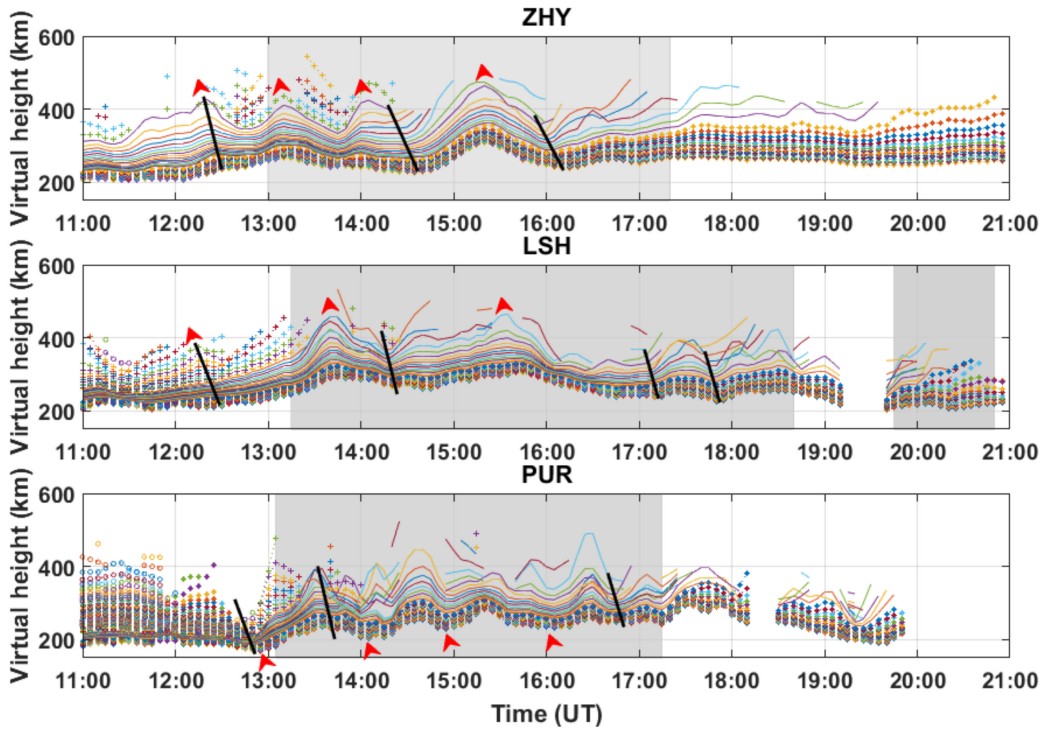

**Figure 6.** The virtual heights at various frequencies (from 2 to 14 MHz with a step of 0.05 MHz) at ZHY, PUR, and PUR stations during 9 September 2017(11:00–21:00 UT). The black lines indicate the downward movement of the phase velocity of traveling ionospheric disturbances (TIDs) in the ionosphere, and gray bars represent the duration of spread F. The red arrows are used to mark the period of TIDs.

Figure 7a shows the time variations of three stations' ROTI on 9 September 2017. Figure 7b shows two-dimensional ROTI maps of ionospheric irregularities with a 10 min time interval from 14:50 UT to 17:20 UT on 9 September 2017. Black stars represent ionosondes located at PUR, LSH, and ZHY stations, respectively. The ROTI value range of the color bar is 0.1–0.3. In general, ROTI magnitudes of >0.25 are considered equatorial plasma bubbles (EPBs) [43]. It can be seen from Figure 7a, the value of ROTI is clearly more than 0.25 between ~15:30 UT and ~16:00 UT at the PUR station (low latitude), and

at the LSH station (low latitude) around 16:40 UT. The value of ROTI at the ZHY station (middle latitude) is lower than 0.25. Correspondingly, in Figure 7b, the plasma bubbles with relatively small-scale ROTI covered a broad area mainly within 20°–35°N and 100°–120°E between 15:00 UT and 16:50 UT. It also can be seen in Figure 7b that there are no plasma bubbles or irregularities near the ZHY station located at middle latitudes, instead of near PUR and LSH stations, which were located at low latitude. Such phenomenon confirms that there are EPBs observed by GNSS over geomagnetic low latitude and equator. The ROTI with relatively small magnitude indicates that the disturbance intensity of irregularities to TEC is not drastic, and the irregularities might be small-scale structures during the storm recovery phase.

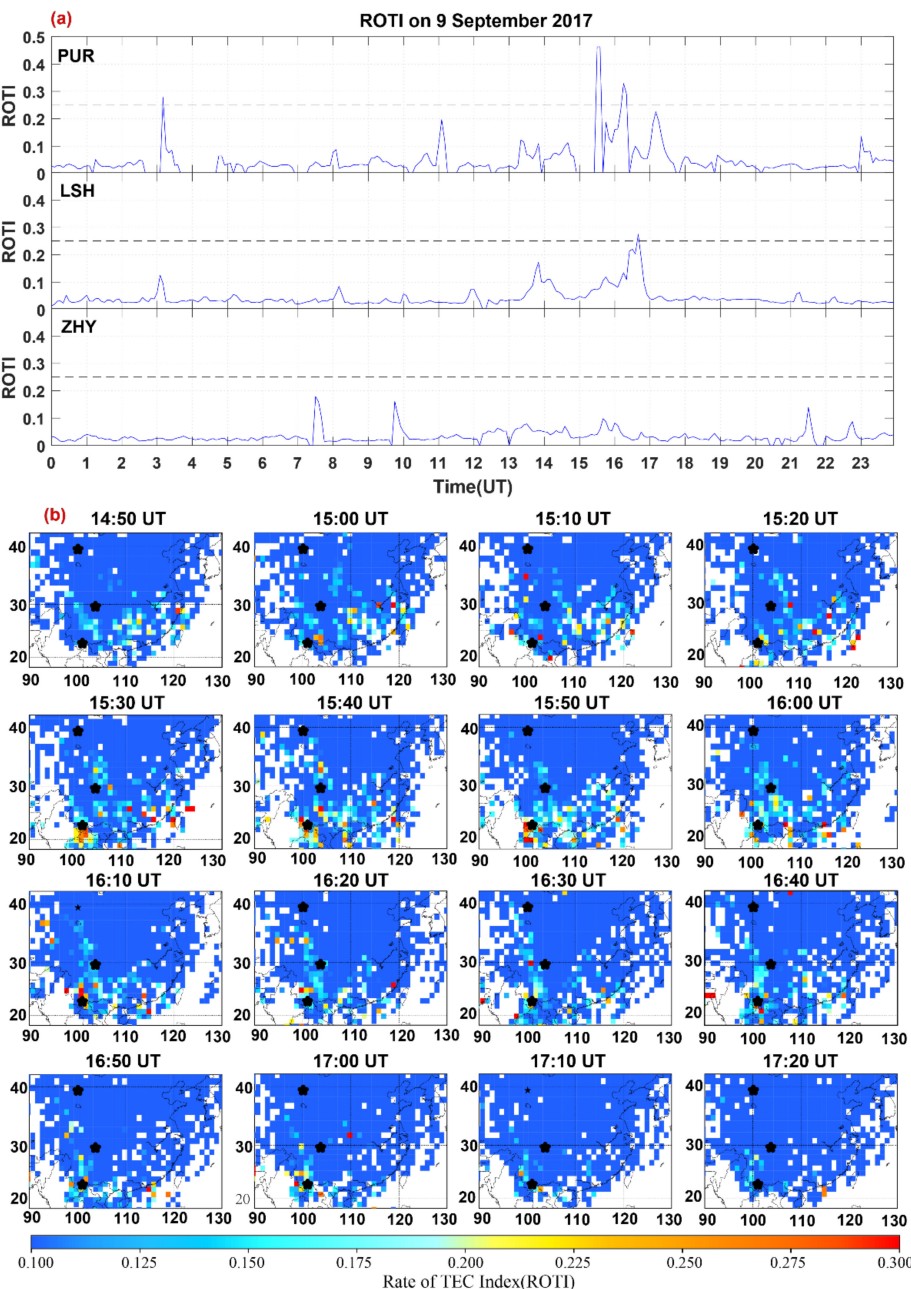

**Figure 7.** (**a**) The time variations of three stations' ROTI on 9 September 2017; (**b**) two-dimensional ROTI maps of ionospheric irregularities in geographic coordinates over the South East Asia sector from 14:50 UT to 17:20 UT on 9 September 2017. Black stars represent the ionosondes located at PUR, LSH, and ZHY stations, respectively.

Figure 8a indicates that only Swarm A passes over China and adjacent sectors in the local nighttime along the longitude of around 110°E between 14:29 UT and 15:00 UT on 9 September 2017. Red triangles show the moving direction of the Swarm A satellite. Figure 8b shows the variation of electron density with respect to latitude along its track recorded by Swarm A. Swarm B did not pass over these stations, and Swarm C has a similar result to Swarm A due to tiny longitudinal deviation and thus is not shown here. According to the formula ((long1 − long2)*110*cos(θ) km) in Yang et al. [44] presented to calculate the vertical distance between Swarm satellites and ionosonde station, it can be worked out that the ground distance between Swarm A and three ionosonde stations are ~913 (PUR), ~574 (LSH) and ~842 km (ZHY), respectively. Due to the fact that the beam width (the angle between the two half-power points of the beam) of the electromagnetic wave passed by the ionosonde is large, the ground distance between Swarm A and the ionosonde is relatively small. Thereby the longitude of around 110°E Swarm A can also be used as a reference study for the ionosphere over the PUR and ZHY stations. Figure 8b shows that topside ionospheric irregularities (fluctuation of electron density) and small-scale scintillations also can be observed at the PUR station (low latitude) at 14:43 UT. However, the fluctuation of electron density of Swarm data at the LSH (14:42 UT) and ZHY (14:39 UT) stations are relatively smooth. That shows that no small-scale scintillation occurred at LSH and ZHY stations between 14:29 UT and 15:00 UT on 9 September 2017.

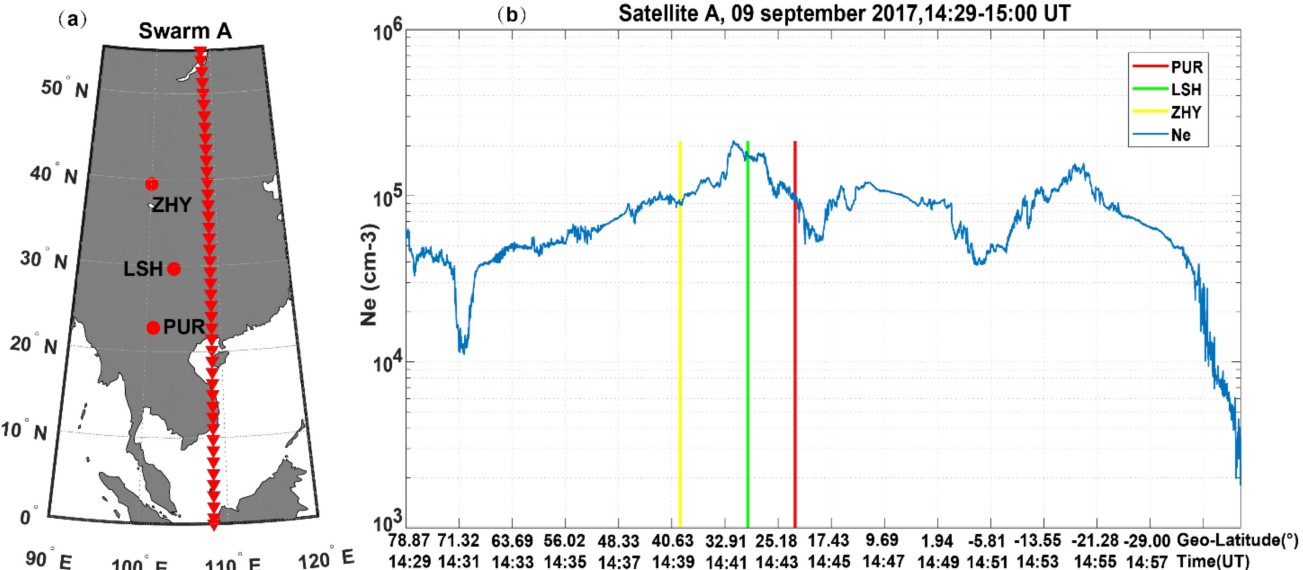

**Figure 8.** (**a**) Geographic map showing the locations of the PUR, LSH, and ZHY ionosondes. Red triangles roughly aligned along 110°E show the moving direction of the Swarm A satellite during the recovery phase of the storm on 9 September 2017; (**b**) the variation of electron density recorded by Swarm A from 14:29 UT to 15:00 UT on 9 September 2017.

## 4. Discussion

As demonstrated by Huang [45] and Abdu et al. [46], the rapid intensification of the AE index, accompanied by a southward turning of the IMF Bz during a geomagnetic storm is a typical signature of the prompt penetration electric field (PPEF) having an eastward polarity. When PPEF is superimposed on the normal prereversal enhancement (PRE) after sunset, it can penetrate into the equatorial ionosphere and then further cause a larger ionospheric layer uplift. This causes an unstable condition in the bottom side of the ionosphere during a magnetic storm [47,48]. This situation is exactly as described in Figure 2. Thus, it is reasonable to think that the eastward prompt penetration electric field superimposed on the normal prereversal enhancement caused the formation of spread F during the second main phase of the storm. In a recent study, Jiang et al. [35] showed this case of large-scale ionospheric irregularities during the second main phase of the storm that

occurred on 8 September 2017. The ionograms pertinent to spread F recorded at PUR, LSH and ZHY stations during the second main phase of the storm are shown in Figures 2–4 of Jiang et al. [35]. More details of spread F during the second main phase of the storm can be read in Jiang et al. [35].

Over this magnetic storm, many researchers [30,32,34,35,49] have investigated the ionospheric irregularities during the second main phase of the storm, while this study has paid more attention to the characteristics of spread F development and evolution during the recovery phase of this storm. Interestingly, it can be seen intuitively from Figure 2 that there are some different phenomena over the existence of spread F between the second main phase of the storm and the recovery phase of the storm. For example, the time sequence of spread F first appearance at three stations is inconsistent at different periods of the storm. The spread F first occurred at PUR (low latitude) during the second main phase of storm, while during the recovery phase, it first appeared at higher latitude regions LSH and ZHY stations, and then occurred at a lower latitude PUR station. Aa et al. [30] suggest that spread F at low and middle latitudes during the second main phase of this storm may be mainly associated with plasma bubbles, because plasma bubbles can be lifted to higher altitudes with depletions being extended along the magnetic field lines to higher latitudes. Thus, the latitudinal progression of spread F in the second main phase is from low to high. However, the latitudinal progression of the spread F during the recovery phase of the storm is likely an effect of the Earth's rotation and the changing solar zenith angle. In addition, the most conspicuous difference is that spread Es could be observed when spread F occurred during the recovery phase of the storm.

Therefore, in order to better probe the ionospheric irregularities during the recovery phase of the storm, Figures 3–5 indicate the onset and development of spread F structures on ionograms at PUR, LSH, and ZHY stations during the recovery phase of this geomagnetic storm on 9 September 2017. As referred from Tsunoda [42], the STs generally exist near the main F layer traces, and sometimes multiple reflections from the bottom side of the F layer also can generate STs. Ionograms in Figures 3–5 show the presence of oblique reflection echoes above the 120 km Es layer. The associated multiple hop traces of the Es layer are also observed on ionograms. Naturally, these oblique traces follow with vertical traces to develop into spread echoes in the aforementioned overlap way. It is intriguing to note that the development process of spread F and spread Es show multiple satellite traces and oblique traces can be observed (see Figures 3–5). Tsunoda [42] suggested that satellite traces sometimes can last for several hours after the ESF. The results presented here strongly suggest the formation of nighttime spread F and spread Es might originate from the same or similar ionospheric perturbations during the recovery phase of the geomagnetic storm.

It is well known that satellite traces (STs) on ionograms are considered important precursors of the large-scale waves (LSWs) in the ionosphere and can be used as indicators of spread F under certain circumstances [33]. Furthermore, Wakai et al. [50] indicated that multiple echoes of Es commonly are regarded as vertically reflected ones from the relative difference of the virtual height. That is typically presented in Figure 5 at 19:45 LT. The Es main trace located at 100 km appears as another trace near 120 km and shows slight spreading after the time of this ionogram. However, the maximum frequency of oblique traces (OTs) exceeds the main trace of Es, even adjacent to the main Es layer trace in ionograms. Thus, oblique traces (OTs) of the Es layer should be considered as the obliquely reflected ones due to the presence of large-scale tilts in the bottom side of the ionosphere.

Bowman [51] suggests there might be a relationship between spread F and spread Es at midlatitudes. Moreover, the author pointed out that the off-vertical reflections of the detection echo induced by the passage of atmosphere gravity waves (AGWs) in the ionosphere might have a significant impact on the formation of spread echoes. Recall from Figures 3–5, multiple off-vertical reflection traces, including satellite traces and oblique traces, are clearly seen. Much of the spreading of ionosonde traces stems from multiple reflections due to modulations of the F region bottom side created by gravity waves. In addition, various studies [33,42,51,52] suggest that satellite traces and multiple reflections

echoes can be observed frequently after sunset. It is considered to be a precursor for large-scale waves (LSWs), and sometimes LSWs play a significant role in the formation of bottom type spread-F (BSF) and ESF. Furthermore, Tsunoda [42] indicated that the large-scale tilts in the bottom side of the ionosphere induced by LSWs might be seeded through the passage of AGWs. Therefore, it is plausible to deduce that AGWs/TIDs might play a more powerful role in forming nighttime spread F and spread Es at low and middle latitudes during the recovery stage of this storm.

To reveal whether TIDs/AGWs exist during the recovery phase of the geomagnetic storm, Figure 6 demonstrates diurnal variations of the virtual height at various frequencies at three stations at 11:00–21:00 UT (18:00–04:00 LT) on 9 September 2017. Frequencies range are from 2 to 14 MHz with a step size of 0.05 MHz. The downward movement of the phase velocity identifies the possible presence of gravity waves in the ionosphere as was reported in many prior studies [53–55]. The result demonstrates that spread F and spread Es during the recovery phase of the storm might be associated with TIDs/AGWs.

It is well known that high-latitude LSTIDs can be triggered by Joule/particle heating in the auroral zone during a geomagnetic storm [56]. In the low- and mid-latitudes regions, the disturbances in the thermosphere/ionosphere sometimes result from a superposition of the lower atmospheric effects and traveling ionospheric/atmospheric disturbances (TIDs/TADs) caused by the high-latitude energy inputs [57]. In the lower atmosphere, the sources of AGWs are various, including mesoscale turbulence and convection, earthquakes, typhoons, the passage of the solar terminator and solar eclipse, meteorological disturbance, and orography effects (mesoscale disturbances due to a stationary incoming stream flows around mountains) [58], and references therein. These sources increase turbulent processes in the lower atmosphere, and then affect the state of the neutral gas in the local atmosphere and ionization-recombination processes of the ionosphere, which cause the change of the parameters in the ionosphere. In the present study, spread F/Es and TIDs/AGWs can be observed in all three stations at middle and low latitude regions. In Figure 6, the periods of TIDs at three stations are approximately 1–1.5 h. Typically, the period of LSTIDs is 1~4 h. According to the period of TIDs, it might be reasonable to think that TIDs/AGWs in the recovery phase of the storm can be classified into LSTIDs. In general, during the geomagnetic storm, the upper atmosphere is heated due to Joule and particle precipitation in the polar region, together with the momentum transfer from electric field drifting ions, leads to the neutral atmospheric gravity waves that appear as LSTIDs [56]. Therefore, TIDs/AGWs might be mainly induced by the high-latitude energy inputs during the recovery phase of this storm. The passage of LSTIDs in low and middle latitudes generates radio waves scattering, shown as spread F and spread Es in Figures 3–5. However, it should be noted that TIDs/AGWs during the recovery phase of the storm might mix multiple scales of TIDs/AGWs in low and middle latitudes, including the residual large-scale TIDs generated in the main phase period [31,59,60], medium-scale and small-scale TIDs originated from the local lower atmosphere, especially at the ZHY station where gravity waves are abundant [61]. During the storm, atmosphere disturbances are violent and often accompanied by atmospheric convection. Thus, LSTIDs mixed with TIDs/AGWs of multiple scales might be the result of interaction between the upper atmosphere and the lower atmosphere. However, our current analysis does not give information on the relative importance of the factor.

Generally speaking, there are two different mechanisms for TIDs/AGWs to generate spread F. The first candidate is to create large-scale tilts or LSWs of the ionosphere [3], which can cause the off-vertical reflections from the ionosphere. The second mechanism is to initiate a local electric field in the F region to generate local ionospheric irregularities (GRT instability at low latitude and Perkins instability at middle latitude). Huang et al. [15] suggest that the large-scale electric fields induced by gravity waves in the E or conjugate F region could connect with the F region by the magnetic field and then cause disturbances of the ionosphere. In addition, Miller [62] suggested that the spatial variations of field line integrated ionospheric conductivity caused by gravity waves can also produce local

electric fields. It is known that the electrodynamic coupling between the E and F regions is significantly responsible for the generation of ionospheric irregularities of E and F layers at low latitudes and middle latitudes [9,17,19,62–64]. The small-scale structures of irregularities in Figures 7 and 8 with an interesting chronological order of the occurrence of spread Es and spread F in Figure 2e–g might implicate that the unstable coupled system of the E-F region was formed by the disturbance electric fields caused by TIDs/AGWs. However, this requires more observations and events to further verify the hypothesis.

## 5. Conclusions

In this study, the evolution and characteristics of spread F and spread Es that occurred in the nighttime low and middle latitudes over China are investigated by using data from ionosondes, Swarm satellites, and GNSS observations during the recovery phase of the 7–9 September 2017 geomagnetic storm. Nighttime spread F recorded by ionosondes during the recovery phase of the storm (9 September 2017) was accompanied by the occurrence of spread Es in the post-sunset sector. The STs and OTs were also recorded on ionograms during the formation and development of the spreading echoes at three ionosondes. Results show that the passage of TIDs/AGWs in the ionosphere and atmosphere during a magnetic storm likely contributes significantly to the creation of the off-vertical reflection echoes and spread traces. During the recovery phase of the storm, the periods of TIDs/AGWs are ~1 h at the ZHY station, ~1.5 h at the LSH station and ~1 h at the PUR station, respectively. Based on the results that the periods of TIDs/AGWs are about 1–1.5 h, TIDs/AGWs might be LSTIDs, which can be induced by the high-latitude energy inputs during the recovery phase of this storm.

**Author Contributions:** Data curation, L.W.; methodology, L.W.; software, C.J.; validation, Y.H., W.H., G.Y., and J.L.; investigation, C.J.; writing—original draft preparation, L.W.; writing—review and editing, C.J., and E.A.; project administration, Z.Z.; funding acquisition, C.J. All authors have read and agreed to the published version of the manuscript.

**Funding:** This research was funded by the National Natural Science Foundation of China (NSFC), grant number 42074184, 41674183, and 41974184, respectively.

**Institutional Review Board Statement:** Not applicable.

**Informed Consent Statement:** Not applicable.

**Data Availability Statement:** The data used in this study are available from Zenodo: https://zenodo.org/record/3972507 (accessed on 4 March 2021).

**Acknowledgments:** We thank the World Data Center (Kyoto) and the International Service of Geomagnetic Indices (ISGI) of the IAGA for providing the Dst and Kp indices, respectively. The Bz index was downloaded from the website www.srl.caltech.edu/ACE/ASC (accessed on 4 March 2021). Dip latitudes and Magnetic latitudes were calculated by IGRF-12 in this study. The Swarm data are provided by the European Space Agency (https://earth.esa.int (accessed on 4 March 2021)). We acknowledge the Institute of Earthquake Forecasting and National Space Science Center for providing ionosonde data. The data used in this study are available from Zenodo: https://zenodo.org/record/3972507 (accessed on 4 March 2021).

**Conflicts of Interest:** The authors declare no conflict of interest.

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
