# Peer review of "Ionosonde Observations of Spread F and Spread Es at Low and Middle Latitudes during the Recovery Phase of the 7–9 September 2017 Geomagnetic Storm"

_remotesensing, doi:10.3390/rs13051010_

Round 1

Reviewer 1 Report

The paper «Ionosonde Observations of Spread F and Spread Es at Low and Middle Latitudes During the Recovery Phase of the 7-9 September 2017 Geomagnetic Storm» is devoted to the study of the behavior of the E and F layers of the ionosphere during the recovery phase of the geomagnetic storm on September 9, 2017. The storm in September 2017 was one of the strongest geomagnetic events in recent years and is undoubtedly of scientific interest. However, this article does not seem to be a complete comprehensive scientific study and makes a rather fragmentary impression.

The authors use several well-known mechanisms for the study, in particular, ionosonde data, SWARM data, and ROTI maps, but there is no agreement on the data of these instruments for the study of a geomagnetic storm.

  1. Figure 2 shows an analysis of the main geomagnetic indices and F2 layer height data for three ionosondes in China. For these ionosondes, the average value for September 2-6 was calculated. Typically, a 27-day average for the critical frequency of the F2 layer is used to study the effects of geomagnetic storms.

The gray lines show the appearance of spread F, but there are also gaps in the data in the figures, which may also correspond to the periods of occurrence of spread F. The authors do not explain the gaps in the data.

  1. Figure 2 also shows the time of occurrence of spread Es but does not show the dynamics of Es behavior during the entire study period, so it is impossible to conclude with certainty that spread Es appears only during the storm recovery period. I would recommend showing the dynamics of Es and explaining in more detail the spread Es and the physical mechanisms of the manifestation of these phenomena.
  2. The ROTI maps are presented only for a short period of time on September 9 and the article does not discuss the effects that are observed in the figure. What did the authors want to show by this? Strong changes in the mouth and not visible.

I would recommend investigating the ROTI for one or more stations over a long period of time. A detailed study of the ROTI during the geomagnetic storm periods can be seen in the papers by Zakharenkova and Cherniak, for example.

  1. SWARM data is also presented for a short time. There is no comparison of these data for calm conditions, the main phase of the storm, and the recovery phase, and therefore it is not clear how the authors conclude about the observed effects from the SWARM satellite data. Similar variations in electron density at the PUR station at 14.43 can also be observed at 14.33, and at 14.58 even larger amplitudes of electron density variations are observed, but this aspect is not discussed by the authors. The physical mechanisms of such effects are also not discussed in detail.
  2. The authors point out that atmospheric gravity waves play a crucial role in the formation of ESF, however, nowhere in the text are the characteristics of these waves given. AGW, indeed, play an important role in the dynamics of the ionosphere. The class of atmospheric gravity waves is very wide and differs in both sources and sizes, methods of propagation, and influence on various layers of the ionosphere. In particular, high-frequency AGW are mainly of tropospheric origin (volcanic eruptions, earthquakes, hurricanes, etc.). Large-scale AGW can indeed be associated with geomagnetic activity. The article does not discuss these aspects in any way.

However, in [Tsunoda, 2008] note that “large‐scale wave structure (LSWS) does not appear to be a simple seeding by atmospheric gravity waves [Tsunoda, 2005]. That is, random seeds appear to be ubiquitous [Eccles, 2004], whereas LSWS and ESF have a day‐to‐day variability. Much remains to be done, but it seems clear that progress in understanding LSWS would be a step in the right direction”.

Based on the above, I would not recommend this article in this version for publication in the journal “Remote sensing”.

Reviewer 2 Report

The paper entitled “Ionosonde Observations of Spread F and Spread Es at Low and

Middle Latitudes During the Recovery Phase of the 7-9 September 2017 Geomagnetic Storm" by Lehui Wei et al. investigate the oscillation in the ionosphere during the recovery phase of 7-9 September 2017 magnetic storm. The paper presents results that are worth to be published. Before publishing, I recommend major revision before the publication.

General Comments

I suggest the authors separate the results of the discussions. Because I got confused in some parts due to a mix of results and discussions.

Specific comments

Lines 306-308: “. Therefore, it might be reasonable to think that TIDs/AGWs might be induced by the high-latitude energy inputs during the recovery phase of this storm.”

The authors suggest that the oscillation comes from high latitudes due to the recovery phase of the magnetic storm. However, I did not see in the fixed frequency plots a phase difference in the three different latitude observations showing the oscillations propagating to equatorward.

Another point, the ionograms do not show a satellite trace sequence for a minimum of 30 minutes. Also, satellite traces appear at almost the same time in ZHY and PUR. This suggests that TIDs are of different origin. The authors need to prove that these oscillations originate at high latitudes due to magnetic storms. For me, it is not clear

item number 3 of the conclusions (lines 388-390) is not a conclusion. Because you did not discuss and did not show results to prove this statement. Therefore, I suggest removing this phrase.

Reviewer 3 Report

Wei et al. analyze observations of the E and F region ionosphere using ionosonde’s located over China at ~100° longitude, Swarm satellite data, and GNSS ROTI to describe the evolution of spread Es and F over the 7-9 September 2017 period. They attribute the occurrence of this phenomenon to TIDs / AGWs induced by the ionospheric plasma flow in response to the changing geomagnetic activity, which produce off-vertical reflections off a tilted ionosphere. They discussion the formation of a polarization electric field in the E region which couples to the F region by magnetic field lines which produces the small-scale irregularities observed by Swarm.

This paper provides an interesting review of the 07-09 September event with a thoroughly referenced introduction to and discussion of the phenomena. However, significant attention to language is required before publication to correct grammar and to remove double-statements. I have made significant annotations in the document to address as much of the grammar as possible. I also have a number of minor comments regarding the comment of the paper itself. During the second review I may have additional comments about the science.

Major revision items:

  1. Language: First, I want to recognize that the authors are not writing in their first language and congratulate them on this accomplishment. The paper is well written and understandable, but requires some attention to bring it to a standard necessary for Remote Sensing. For the most part the necessary corrections are minor and have been included in the annotated pdf included with this review. However, there are a number of places where I was unable to understand the sentences. There are also several instances where the authors repeat themselves or have very long sentences or paragraphs which need to be split up.
  2. Solar source (e.g., line 90, line 160): Be careful in what you attribute the solar source of the ionospheric and geomagnetic activity. As written, the authors describe the X9.3 solar X-ray flare on September 06th as the source of the observed activity. However, the solar X-ray flare only increased photoionization on the dayside of the Earth for a relatively short period on September 6th. The geomagnetic and ionospheric disturbances described in this paper can be attributed to energetic electron precipitation caused by the arrival of energetic electrons with the CME associated with the flare. The flare itself is not the solar source, as written, the CME is. Note also that the solar flare did not cause the CME, but they did originate from the same active region.
  3. Role of SSC (lines 164-165): There is a cause and effect error in the descriptions of the SSC. As written, the SSC is said to cause the northward turning of the IMF. However, this simply is not true. The SSC is a result of a sudden change (shock) in solar wind parameters such as speed, density, or pressure. An SSC is a feature of the geomagnetic field and does cause the IMF Bz to change. Rather the SSC is caused by changes in the solar wind and IMF. Furthermore, why not reference the time of the SSC? Based on the OBSEBRE database, SSCs associated with this storm were observed on 20170906 at 23:43 UT and 20170907 at 23:00 UT. How do these times fit in with your analysis? [See http://www.obsebre.es/php/geomagnetisme/vrapides/ssc_2017_d.txt].
  4. Section 3, paragraph 1. This is a long paragraph packed with a lot of information. I suggest dividing the paragraph into 3 smaller paragraphs: (1) introduction to the Figure, (2) description of the upper plots and storm phases), (3) description of the lower figures.

Minor revisions are included in the annotated PDF review. In addition, please consider the following:

  1. Where possible I recommend the authors avoid using the work ‘we’ as it is less formal.
  2. Line 158: The referenced plots are indicative of stormtime conditions, not background conditions. Please be more clear.
  3. Line 158: The terms in the brackets is not a list of geomagnetic indices as IMF Bz is included in the list. Please correct the description.
  4. Line 176-177: Suggest describing the times of these peaks. The >400 peak at PUR shortly after the 3 referenced peaks stands out and it should be stated why it is not considered the ‘peak’.
  5. Figure 2: Please increase the font size. I suggest using vertical lines or horizontal bars to clearly indicate the phases of both storms
  6. Line 192: IMF Bz is not a geomagnetic field index.
  7. Lines 207-209: Is the latitudinal progression of the spread F reported in the literature? This is likely an effect of the Earth’s rotation and the changing solar zenith angle.
  8. Figures 3-5: To improve visibility, consider changing the colour scale of the plot. Suggestions include using grayscale and omitting 0 values, or perhaps values <1. The magenta traces are very fine and difficult to make out. I also suggest increasing your font size and the width of the arrows.
  9. Figure 7 discussion (lines 314-324): I found the details in the text very difficult to derive from the Figures. As a non-expert in using such plots, all I see is a uniformly coloured plot. I must assume that the authors are referring to small coloured pixels that seem to show up in some of the plots. I strongly suggest increasing the size of the plots, perhaps by focusing on a smaller region, to better demonstrate the features you are trying to describe.
  10. Figure 8: Text size is too small. Indicate the direction the Swarm satellite is moving.
  11. Conclusion 1: This isn’t a conclusion, this is a statement about what you did.

Round 2

Reviewer 1 Report

Thank you for making the changes in the article. The article has become more understandable, however, I still have some questions.

1. There are still concerns that the effects that you associate with the recovery phase of the storm may manifest on other days. You give the results of observations only on September 9, 2017 for a short time on this storm, which can not judge the effects of only the recovery phase of the storm, because many factors affect the Earth's ionosphere.

The authors on the basis of one study claim about the «morphological characteristics», which I think is not completely true (line 319).

2. The article does not give enough description and discussion about the TIDs/AGWs. 

Indeed, atmospheric gravitational waves have a great influence on the dynamics of the F and E layers of the ionosphere, but most of the AGWs sources are located in the lower layers of the atmosphere, which, in my opinion, is not very correct to associate with the recovery phase of the storm. But it is true that during both the main and the recovery phases of storms, LSTIDs occur, which is described in detail in [53] to which the authors refer.

For example, the solar terminator can be a source of AGW and influence the parameters of the ionosphere.

I сonsider that the statements in the article about TIDs/AGWs should be justified and described in more detail, and additional links have been added to confirm this statement.

Reviewer 2 Report

The paper entitled “Ionosonde Observations of Spread F and Spread Es at Low and Middle Latitudes During the Recovery Phase of the 7-9 September 2017 Geomagnetic Storm" by Lehui Wei et al. investigates the ionospheric irregularities during the recovery phase of the 7-9 September 2017 geomagnetic storm. There are still points to be addressed and I recommend minor revision prior to the publication.

Specific comments

I suggest the authors comment on how this average was made in Figure 2? What was the methodology used and cite references

Lines 211-213: I suggest citing papers that also work with satellite traces.

Line 285: I suggest showing the equation number in Yang et al.

Lines 288-290: “Due to the bean width of the electromagnetic wave passed by the ionosonde is large, the ground distance between Swarm A and ionosonde is relatively small.” What is the beamwidth, specifically?

In discussions about the sources and types of TIDs that generated irregularities, I cannot understand because there is a mix of ideas. In lines 389-396, the authors argue that the TIDs have different spatial scales and that they originate at the lower atmosphere. However, in lines 388-389, the authors claim that they are due to the energy penetration from high latitudes.

The authors are speculating in lines 388-396. Therefore, authors cannot write anything about it because it has no results to support these ideas.

Between lines 397 and 413, the authors need to introduce a sentence informing the reader that TIDs generate ionogram scattering due to 2 types of MSTIDs.

In lines 427 and 428, the sentence is not complete.

Lines 434-436, how is the magnetic storm the source of these TIDs? if AGW / TIDs are generated by convection. Actually, the authors do not know the origin of TIDs, so I suggest removing the phrase from the conclusion.

Lines 443-446: Item 6 is not a conclusion because the authors did not show results that support these arguments. Remove it, please
